

# Mental illness detection through harvesting social media: a comprehensive literature review

Shahid Munir Shah[1], Mahmoud Mohammad Aljawarneh[2], Muhammad Aamer Saleem[1] and Mahmoud Saleh Jawarneh[2]

[1] Faculty of Engineering Sciences and Technology, Hamdard University, Karachi, Pakistan
[2] Faculty of Information Technology, Applied Science Private University, Amman, Jordan

Corresponding author
Shahid Munir Shah,
shahidmunirshah@yahoo.com

## ABSTRACT

Mental illness is a common disease that at its extremes leads to personal and societal suffering. A complicated multi-factorial disease, mental illness is influenced by a number of socioeconomic and clinical factors, including individual risk factors. Traditionally, approaches relying on personal interviews and filling out questionnaires have been employed to diagnose mental illness; however, these manual procedures have been found to be frequently prone to errors and unable to reliably identify individuals with mental illness. Fortunately, people with mental illnesses frequently express their ailments on social media, making it possible to more precisely identify mental disease by harvesting their social media posts. This study offers a thorough analysis of how to identify mental illnesses (more specifically, depression) from users' social media data. Along with the explanation of data acquisition, preprocessing, feature extraction, and classification techniques, the most recent published literature is presented to give the readers a thorough understanding of the subject. Since, in the recent past, the majority of the relevant scientific community has focused on using machine learning (ML) and deep learning (DL) models to identify mental illness, so the review also focuses on these techniques and along with their detail, their critical analysis is presented. More than 100 DL, ML, and natural language processing (NLP) based models developed for mental illness in the recent past have been reviewed, and their technical contributions and strengths are discussed. There exist multiple review studies, however, discussing extensive recent literature along with the complete road map on how to design a mental illness detection system using social media data and ML and DL classification methods is limited. The review also includes detail on how a dataset may be acquired from social media platforms, how it is preprocessed, and features are extracted from it to employ for mental illness detection. Hence, we anticipate that this review will help readers learn more and give them a comprehensive road map for identifying mental illnesses using users' social media data.

## INTRODUCTION

In today's modern world where human comfort has reached to a level better than ever before (through the advent of better and reliable technologies), mental diseases/disorders have also grown to alarming levels. Mental illness is now a global issue affecting the major population of the world (*Otu, Charles & Yaya, 2020*). According to 2019 statistics, out of 2,516 million population, 293 million individuals of age 5 to 24 years were affected by at least one mental disorder including depression, anxiety, bipolar, and schizophrenia *etc.* (*Kieling et al., 2024*). The issue is more severe and prevalent in low-and middle-income countries (*Pedersen et al., 2019*; *De Castro, Cappa & Madans, 2023*), where the failing economy further causes deterioration in peoples' mental health (*Ridley et al., 2020*). COVID-19 has also played a significant role in worsening peoples' mental health conditions (*Cullen, Gulati & Kelly, 2020*; *Blanchflower & Bryson, 2022*).

Mental illness is a precarious disease that at its extremes leads to violent and aggressive behaviors (*Whiting, Lichtenstein & Fazel, 2021*). Such aggressive behaviors make mentally ill individuals dangerous to both themselves and others. One of the extreme aggressive behaviors exhibited by mentally ill people is suicidal attempts (*Favril et al., 2020*). The World Health Organization (WHO) declares mental illness the main cause of suicides that ruins millions of precious lives every year (*World Health Organization, 2019*).

Depression is one of the most prevalent mental disorder that affects more than 300 million people globally (*Safa, Edalatpanah & Sorourkhah, 2023*). In addition, those with depression have a higher mortality risk from suicide than those with other mental illnesses (*Baldessarini, Vázquez & Tondo, 2020*). Early diagnosis of depression is highly advised (before the condition becomes complicated), yet the majority of those with depressive disorders do not seek medical attention or attend clinics at the beginning of their illness. As a result, their condition worsens and they suffer serious health repercussions (*Shen et al., 2017*). Another reason of depressive disorder complications is its difficult diagnostic procedure. Traditionally, depression is diagnosed using self-reports, questionnaires and testimony from friends and relatives (*Smith, Renshaw & Bilello, 2013*; *Paykel, 2022*). Such reports and questionnaires are then analyzed by the experienced psychiatrists. Most of the time, people with depressive disorders do not provide accurate information and even experienced psychiatrists cannot accurately diagnose their disease. Hence, such traditional approaches are mostly unreliable and lead to incorrect results and diagnosis (*Hussain et al., 2020a*).

Fortunately, people who suffer from depressive disorders or the other mental health issues tend to reveal about their condition (what they feel or struggle daily) on the internet over different social media platforms (Twitter, Facebook, WhatsApp, Instagram, Reddit *etc.*) (*Ameer et al., 2022*). Daily activities, status updates, and social interactions of depressed people on social networks has offered researchers a new mode of study. It is now easy for them to mine and analyze users' behavioral patterns through examining their activities on social networks. Instead of traditional approaches to diagnose mental illness, social media content of users' is now widely used to detect mental health issues (*Chancellor & De Choudhury, 2020*; *D'Alfonso, 2020*; *Naslund et al., 2020*; *Zogan et al., 2022*; *Özçelik &*

*Altan, 2023*). In literature, many approaches have been used to detect mental illness and user mood through social media. This includes different Natural Language Processing (NLP) and Text Mining (TM) approaches such as Principle Component Analysis (PCA) (*Amanat et al., 2022*), Linguistic Inquiry and Word Count (LIWC) (*Tausczik & Pennebaker, 2010*), Term Frequency Inverse Document Frequency (TF-IDF) (*Dalal, Jain & Dave, 2023*), Word2vec (*Akhtar et al., 2018*), Global Vectors for word representation (GloVe) (*Kanwal et al., 2019*), FastText (*Selva Birunda & Kanniga Devi, 2021*), and latent Dirichlet allocation (LDA) (*Tong et al., 2022*), machine learning (ML) techniques like support vector machines (SVM) (*Li et al., 2019*), decision trees (DT) (*Yang et al., 2016*) (along with boosting methods such as XGboost), Random Forest (RF) (*Vasha et al., 2023*), naive Bayes (NB) (*Deshpande & Rao, 2017*), K nearest neighbors (KNN) (*Alqazzaz et al., 2023*), Logistic Regression (LR) (*Shen et al., 2017*), Neural Network (NN) and DL architectures such as Multilayer Perceptron (MLP) (*Javed et al., 2021*), convolutional neural network (CNN) (*Choudhary et al., 2024*), recurrent neural network (RNN) (*Tommasel et al., 2021*), long-short-term memory (LSTM) (*Amanat et al., 2022*), bidirectional long-short term memory (BiLSTM) (*Fudholi, 2024*), and transfer learning (TL) (*Ameer et al., 2022*) and its different variations like Bidirectional Encoder Representations from Transformers (BERT) (*Gorai & Shaw, 2024*), and Universal Sentence Encoder (USE) (*Baghdadi et al., 2022*) *etc.*

There exist number of reviews and surveys that provide detail on the mental illness detection using social media data; however, these reviews or surveys lake in multidimensional explanation of the topic and mostly covers the specific details. For example, *Skaik & Inkpen (2020)* in their review, mostly focused on the data collection methods and classification techniques. *Ríssola, Losada & Crestani (2021)* in their survey, focused on the analysis of the computational methods for automatic mental illness detection. *Graham et al. (2019)*, *Su et al. (2020)* in their review, mostly focused on DL techniques. *Iyortsuun et al. (2023)* provided a limited study on discussing ML and DL models, however without providing detail on different preprocessing and feature extraction techniques. *Khoo et al. (2024)* focused their review towards multimodal detection of mental health disorders. A very limited studies focused on specific mental illness like depression detection (*Giuntini et al., 2020*) and to the best of our knowledge, no existing review has provided detailed description of the mental illness detection through social media platforms by explaining data acquisition methods, preprocessing methods, feature extraction techniques, ML, DL, and NLP techniques along with their detailed discussion and analysis.

This review provides a comprehensive literature on mental illness detection (specially depression detection) using users' social media data through NLP, ML, and DL techniques. Important feature extraction methods along with preprocessing techniques have also been discussed in detail. Ideally, this study provides a summary of current research progress and an outlook of the future work. Following are the contributions of this research:

- This review provides a state of the art literature on mental illness detection (more specifically depression detection) through harvesting users' social media data.

- It covers more than 100 recent related literature, which, to the best of our knowledge have not been covered in any other review.
- It includes state of the art ML, DL, and NLP based modelling techniques employed for mental illness detection in recent past along with the popular features extraction techniques.
- It provides a detailed analysis of ML, DL, and NLP techniques along with their strengths and limitations to provide the readers a better idea to use them for their applications.

The remaining of the paper is organized as follows: 'Extensive Literature Review' provides an extensive summary of the research that has been done on mental illness detection in recent past. 'Methodology' explains the complete methodology of designing mental illness detection systems using social media data. It outlines data acquisition methods, preprocessing techniques, features extraction, and classification methods employed to classify mental illness detection. 'Current Challenges and Future Directions' provided the summary of the current challenges and future directions in this specific field and finally, 'Conclusion' provides the conclusion of the study.

# EXTENSIVE LITERATURE REVIEW

In last few decades, an extensive research has been carried out on the task of mental illness detection, specifically using social media data. For this purpose, different techniques have been explored for data collection and features extraction as well as various modeling techniques have been employed to automatically detect and classify various mental illness conditions. In order to review the most recent literature in this specific area, a methodology has been followed so that an unbiased coverage of the literature is achieved.

## Survey methodology

The methodology opted in this review includes article selection, article filtration, and data extraction processes. Each of these processes is explained below.

## Article selection process

The articles for this literature review have been collected through searching from several important databases including Science Direct, Springer Link, Wiley, IEEE Xplore, ACM Digital Library, Scopus, PubMed, Web of Science, and Google Scholars (see Fig. 1 for reference). A search query based on six keywords *i.e.,* ''Mental Illness Detection'', ''Depression Detection'', ''Mental Illness Classification'', ''Mental Illness Detection Using Social Media Data'', ''Depression Detection using Social Media Data'', ''Mental Illness Classification using Social Media Data'' was used to retrieve the articles from the databases. Keywords were used individually or after combining them through Boolean operator ''OR''. Different combinations of keywords have been shown in Table 1. Keywords and set of keywords were searched in the title or abstract depending on the search options given by the databases.

Table 2 provides the summary of the articles retrieved from each database using the keywords/sets of keywords.

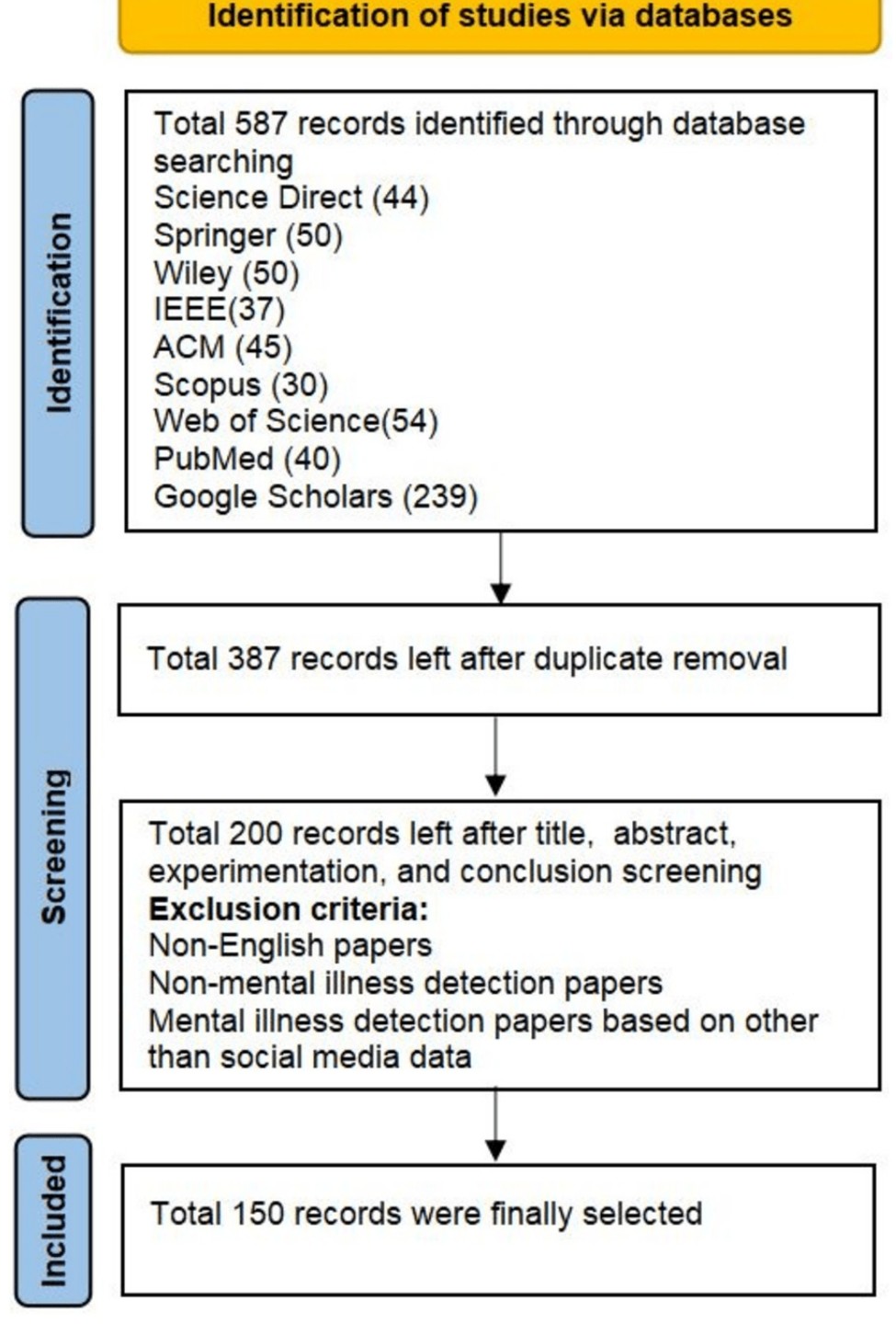

**Figure 1  Literature review methodology.**

**Table 1  Keywords and their combinations for search query.**

| Set of keywords | Category | Combined keywords |
|---|---|---|
| Set-1 | Mental Illness | "Mental Illness Detection" OR "Mental Illness Classification |
| Set-2 | Mental Illness | "Mental Illness Detection Using Social Media Data" OR "Mental Illness Classification using Social Media Data" |
| Set-3 | Depression | "Depression Detection" |
| Set-4 | Depression | "Depression Detection using Social Media Data" |

**Table 2  Articles review statistics.**

| Keywords | Science direct | Springer | Wiley | IEEE | ACM | Scopus | Web of science | Pub med | Google scholars | Total |
|---|---|---|---|---|---|---|---|---|---|---|
| Set-1 | 8 | 11 | 05 | 6 | 07 | 07 | 11 | 05 | 65 | 125 |
| Set-2 | 06 | 07 | 05 | 08 | 05 | 07 | 05 | 05 | 44 | 92 |
| Set-3 | 18 | 20 | 15 | 12 | 20 | 07 | 11 | 26 | 70 | 199 |
| Set-4 | 12 | 12 | 15 | 11 | 13 | 09 | 27 | 14 | 60 | 173 |
| Total articles retrieved | 44 | 50 | 40 | 37 | 45 | 30 | 54 | 50 | 239 | 589 |

## Articles filtration process

As shown in Table 2, a total of 589 articles have been retrieved from the selected databases. After the articles have been retrieved, the next process is the inclusion-exclusion criteria. For this purpose, the retrieved literature is processed from different filters to achieve the final literature data. The initial filter is to exclude duplicate articles and a total of 387 records left after the duplicate removal. The remaining articles were further screened based on reviewing their abstract, introduction, experimentation, and conclusion. Following exclusion criteria was used: (1) the article was not available in English (2) the abstract and conclusion were not relevant to mental illness detection, and (3) the abstract was relevant to mental illness but the papers did not use the social media data instead used other methods for data collection. Finally, after all this screening, 150 articles were left and were processed for review. This way, only those papers were included in the final dataset, which were the most relevant to our study. We included publications up to May 2024 in our literature review. The flow diagram of the article selection process is shown in Fig. 1.

## Data extraction

From each of the retrieved articles, the following data is extracted:

Year of Publication, Data Source, Features Extracted, Classification Techniques used, Unique Contributions, and Limitations. Table 3 provides the detail of the extracted data from the retrieved articles.

## METHODOLOGY

This section provides the detail of the complete process of designing mental illness detection systems using social media data. It includes data acquisition, data preprocessing, features

**Table 3  Summary of literature review.**

| Reference | Disease | Data source | Features extracted | Classifier(s) used | Contributions | Limitations |
|---|---|---|---|---|---|---|
| Oliveira & Paraboni (2024) | Depression, anxiety | Twitter | BoW, LIWC | BERT | The proposed model achieved 0.40 F1-score for depression prediction and 0.36 F1-score for Anxiety prediction, | The study only focused on Portuguese language. |
| Gorai & Shaw (2024) | Suicide risk | Twitter, Reddit | N-A | BERT, CNN | The proposed model performed better as compared to the recent approaches in detecting suicide risk. | The study only focused on suicide risk prediction. |
| Fudholi (2024) | Border line personality disorder (BPD), anxiety, depression, bipolar, mental illness, schizophrenia, and poison | Reddit | FastText | CNN-BiLSTM | Modeling with CNN-BiLSTM and FastText embedding provided an F1-score and accuracy of 85.0% and 85.0%, respectively. In comparison to the BiLSTM model, the F1-Score and accuracy were both 83.0%. | The study used a comparatively smaller dataset i.e., 35,000 training records and 6,108 test records. |
| Alqazzaz et al. (2023) | Depression | Twitter | Keywords extraction | LR, KNN, SVM, and CNN-LSTM | CNN-LSTM achieved a superior result i.e., 86.2% detection accuracy as compared to the other employed techniques. | The study used a comparatively smaller dataset i.e., 36,587 tweets. |
| Vasha et al. (2023) | Depression | Facebook, YouTube | TF-IDF | NB, SVM, RF, DT, LR, KNN | Among the used classifiers, SVM achieved the highest accuracy i.e., 75.15%. | The study used a comparatively smaller dataset (10,000 posts). Furthermore, ML models were used for depression detection, however more recent DL methods could be used. |
| Banna et al. (2023) | Depression | Twitter | Word2vec | CNN, LSTM | The employed models achieved overall 99.4% detection accuracy. | Social media posts between October 2019 and May 2020 were used. Furthermore, the number of participants were limited i.e., 400, and the study was confined to the four major cities of UK only. |
| Lamichhane (2023) | Depression, stress, suicidality | Reddit | N-A | ChatGPT | ChatGPT API classified the social media posts and obtained F1-score of 0.73, 0.86, and 0.37 for stress, depression, and suicidality detection, respectively. | GPT-3.5-turbo backend was used, however, much improved backend of ChatGPT was released and the model could be tested on that. Furthermore, during the model evaluation, a limited prompt setting of the ChatGPT was explored and the first response from the ChatGPT was considered as its prediction, There might be other prompts that can provide better classification. |
| Dalal, Jain & Dave (2023) | Depression | Twitter, Reddit | TF-IDF, Word2vec | SVM, NB, NN, CNN, LSTM, BiLSTM | SVM outperformed the other classifiers and achieved the highest detection accuracy i.e., on average upto 70.0%. | A dataset was build without the help of the domain experts. Furthermore, the size of the dataset was limited for the better evaluation of DL models. |
| Amanat et al. (2022) | Depression | Twitter | PCA, One-Hot, TF-IDF | RNN-LSTM | Proposed RNN-LSTM framework achieved 90.0% detection accuracy. | The dataset was limited for the better evaluation of DL models. |
| Ameer et al. (2022) | Depression, anxiety, bipolar disorder (BD), attention deficit hyperactivity disorder (ADHD), and post-traumatic stress disorder (PTSD) | Reddit | N-gram with the highest TF-IDF values | ML (RF, Linear SVM, Multinomial NB, and LR), DL (CNN, Gated Recurrent Unit (GRU), Bidirectional Gated Recurrent Units (Bi-GRU), LSTM, and BiLSTM), and TL (BERT, XLNet) | Pre-trained TL RoBERTa model outperformed the other traditional ML and DL algorithms with an accuracy score of 0.80%. | A multi class dataset was used to detect several mental illness conditions, however, it could be more effective to use a multi label dataset because a post can have more than one mental disease instead of one per post, i.e., depression, Anxiety etc. |

**Table 3** (*continued*)

| Reference | Disease | Data source | Features extracted | Classifier(s) used | Contributions | Limitations |
|---|---|---|---|---|---|---|
| *Wani et al. (2022)* | Depression | Facebook, Twitter, YouTube | Word2vec, TF-IDF | CNN, LSTM | Word2vec LSTM and Word2vec (CNN+LSTM) models achieved 99.02% and 99.01% detection accuracies, respectively. | A large dataset was developed by merging three existing datasets collected from Facebook, Twitter, and YouTube. This increased the complexity and biases in the dataset. |
| *Figuerêdo, Maia & Calumby (2022)* | Depression | Reddit | Context-Independent Word Embedding (CIWE) and Early and Late Fusion Approaches | CNN | The proposed system achieved a precision of 0.76 with equivalent or superior effectiveness in relation to many kbaselines word embedding based approaches. | CNN in combination with Early and Late Fusion was used, however, more novel techniques along with CNN could be investigated. |
| *Baghdadi et al. (2022)* | Suicide | Twitter | Word2vec and GloVe | BERT, USE | The trained USE models achieved the best Weighted Sum Metric (WSM) at 80.20%, and BERT models have achieved the best WSM at 95.26%. | Arabic tweets were used for suicide detection, however, multiple mental illness conditions could be detected through the employed dataset. |
| *Almars (2022)* | Depression | Twitter | Word2ve | BiLSTM | An attention mechanism is combined with a BiLSTM for depression detection. The proposed model achieved 0.83% detection accuracy. | A Twitter dataset of approximately 6,000 tweets was used, however, for the better evaluation, a large scale dataset may be required with DL models. |
| *Safa, Bayat & Moghtader (2022)* | Depression | Twitter | N-gram, LIWC, Automatic Image Tagging, and Bag-of-Visual-Words | Multimodal analysis | Tweets and bio-text showed 91.0% and 83.0% accuracies in predicting depressive symptoms, respectively. | Only self reported tweets were used to detect depression. In general using mixed tweets (self-reported and random) based on generic depression related keywords may be useful. |
| *Tong et al. (2022)* | Depression | Twitter | LDA | Cost-sensitive Boosting Pruning Trees (CBPT) | In order to boost the training process, a resampling weighted pruning algorithm was combined with CBPT. The proposed framework effectively identified depressed Twitter users with 88.39% accuracy. | CBPT method was compared with several tree based methods, however, more recent DL methods could be compared for the better comparative analysis. |
| *Alabdulkreem (2021)* | Depression | Twitter | Word2vec and GloVe | LSTM | The proposed LSTM model effectively detected depressed women with 76.0% detection accuracy. | Depression was detected in Arab women through their tweets, however, for the unbiased depression detection, data should be used of both genders *i.e.*, men and women. Furthermore, the study was limited to COVID-19 period only. |
| *Bae, Shim & Lee (2021)* | Schizophrenia | Reddit | LIWC | SVM, LR, NB, RF | ML algorithms distinguished schizophrenic from control posts (non-mental health related posts (fitness, jokes, meditation, parenting, relationships, and teaching) for the control group) with 96.0% accuracy. | Traditional ML methods were used, however, more recent DL algorithms could be used for better comparative analysis. |
| *Chiu et al. (2021)* | Depression | Instagram | Word2vec | CNN | Considering the time interval between posts, a two-stage detection mechanism was used for detecting depressive users. The proposed methods achieved overall 0.835 F1-score for detecting depressive users. | Tweets collected from 520 users only, however, analysis on a large scale datasets could be more attractive for a multimodal system. |
| *Tommasel et al. (2021)* | Mental illness (anxiety, depression, stress), emotions (negative, positive) | Twitter | FastText | RNN | RNN was trained on captured lexicons through FastText to classify different emotions and mental health conditions. Results showed that RNN were capable of accurately identifying emotions and mental health disorders. | The study focused on only COVID-19 pandemic period and Argentina region. |
| *Kim et al. (2020)* | Depression, anxiety, bipolar, BPD, schizophrenia, autism | Reddit | TF-IDF | XGBoost, CNN | Six independent models (one for each mental disorder) were trained. Each trained models efficiently identified people with that particular mental diseases for which the model was trained, however, in all the models, CNN based models outperformed the XGBoost based models. | The employed models were trained on a specific mental state to directly classify the symptoms and provide the predicted probabilities for each symptom. In this way, it could not accurately measure the co-morbid mental illness status. |

extraction, and classification steps. Detail of each step along with their critical analysis is provide below.

## Data acquisition

Data acquisition is an essential component of detection and classification through learning algorithms. For this purpose, initially, the data is crawled from social media platforms and then important features of it are used to train the algorithms. In literature, different procedures have been adopted to crawl data from social media platforms; however, in all the methods, as a common practice, a particular application programming interface (API) of each social media platform is used to crawl the data. Data collection mechanisms using social media API are often loosely controlled, resulting in out-of-range values, impossible data combinations, missing values, *etc.* After collection of data using API, the first observation is the variations in the raw data (unorganized data) that often lead to a difficult features extraction process. The raw data effects the efficiency of extracting reliable features because it is hard to find word similarities in it and apply semantic analysis to it. Therefore, before features extraction, raw data must be preprocessed to make it suitable for features extraction process. Table 4 provides the details of the datasets crawled from social media platforms in the reviewed literature.

## Data preprocessing

Data preprocessing is a process to prepare the extracted text data to make it suitable for features extraction process. Several preprocessing steps are followed for this purpose, detail is as follows.

### Tokenization

Tokenization is the initial activity of any text processing application. It is a method of dividing the text material into smaller pieces known as tokens. The token could consist of words, phrases, numbers, symbols, or other significant components. The tokens serve as an input for additional preprocessing tasks (*Hussain et al., 2020b*).

### Stop words removal

Stop words are used to divide natural language. These include articles, prepositions, and pro-nouns, *etc.*, which do not add any meaningful information to a document. The purpose of removing stop words from the text is to make language simpler and to only include meaningful information for the further processing (*Resnik et al., 2015*; *Deshpande & Rao, 2017*; *Kumar, Sharma & Arora, 2019*). Because these words are not measured as keywords in text mining applications, therefore, their removal do not produce any affect in certain applications like information retrieval (IR), auto-tag generation, caption generation and several text classification tasks like spam filtering, language classification, and mental illness classification using text data *etc.* However, removing stop words is not preferable in some applications like machine translation, language modelling, text summarization, sentiment analysis, *etc.* where removing stop words may result in misinterpretations of the meanings (*Ladani & Desai, 2020*).

**Table 4  Data acquisition in the reviewed literature.**

| Reference | Dataset description |
| --- | --- |
| *Oliveira & Paraboni (2024)* | 46.8 million tweets written in Portuguese by 18,819 unique users, and their set of friends, followers, and mentions were collected from Twitter. |
| *Gorai & Shaw (2024)* | Dataset was collected from Twitter and Reddit (1) A total of 10,314 tweets, including 2,314 labeled as depressed and 8,000 labeled as non-depressed plus 1,747 sentences from CEASEv2 (*Ghosh, Ekbal & Bhattacharyya, 2022*) were combined with 2,314 depressed tweets (2) 1,916 sucidal risks related posts from Reddit were collected and 1,747 sentences from CEASEv2 were added in it. Furthermore, 2,727 neutral (non-suicidal) Reddit posts were also collected. |
| *Fudholi (2024)* | 41,108 samples including 35,000 training samples, and 6,108 test samples were collected from Reddit. |
| *Alqazzaz et al. (2023)* | 36,587 samples were collected from Twitter users, and their set of friends, followers, and mentions. |
| *Vasha et al. (2023)* | 10,000 Facebook posts and YouTube comments have been collected and split into 80% and 20% ratio for training and test data, respectively. |
| *Banna et al. (2023)* | Tweets from two datasets have been combined and used *i.e.*, (1) 8,000 tweets from sentiment140 *Go, Bhayani & Huang (2009)*, and (2) 2,345 tweets from Depressive_Tweets_Processed dataset. |
| *Lamichhane (2023)* | A total of 3,553 Reddit posts split into 2,838 training and 715 test posts were collected. |
| *Dalal, Jain & Dave (2023)* | A total of 9,000 posts from Twitter and Reddit were collected from depressed and control users. |
| *Amanat et al. (2022)* | More than 4,000 depressed and non-depressed tweets were collected from Twitter. |
| *Ameer et al. (2022)* | A total of 16,930 posts, divided into training (13,726), development (1,716), and test (1,488) posts, were collected from Reddit. |
| *Wani et al. (2022)* | A Twitter dataset with 11,590 tweets (depressive (6,080), non-depressive (5,510)), a Facebook dataset with 5,700 posts (depressive (2,700), non-depressive (3,000)), and a YouTube dataset with 14,029 posts (depressive (7,520), nondepressive (6,509)), were used. |
| *Figuerêdo, Maia & Calumby (2022)* | A total of 2,000 Reddit posts were collected from 887 users including 135 depressive and 752 nondepressive users. |
| *Baghdadi et al. (2022)* | A total of 14,576 suicide related tweets were collected from Twitter between 01-10-2017 and 20-01-2022. |
| *Almars (2022)* | A total of 6,000 Arabic language tweets related to depression were collected from Twitter between February and the April, 2021. |
| *Safa, Bayat & Moghtader (2022)* | A total of 1,1890,632 tweets were collected with random depressive and non-depressive posts from Twitter. |

**Table 4** (*continued*)

| Reference | Dataset description |
| --- | --- |
| *Tong et al. (2022)* | Two datasets were used *i.e.*, (1) Tsinghua Twitter Depression Dataset (TTDD) *Shen et al. (2017)* and (2) the CLPsych2015 Twitter Dataset (CLPsych2015) (*Coppersmith et al., 2015*). TTDD contains 66,672 tweets, while, CLPsych2015 contains 3,000 tweets per users with 477 depressed, 396 PTSD, and 873 control users. |
| *Alabdulkreem (2021)* | A total of 10,000 tweets were collected from 200 women Twitter users during the COVID-19 pandemic period. |
| *Bae, Shim & Lee (2021)* | 60,009 schizophrenia related posts from 16,462 Reddit users and 425,341 controlled posts from 248,934 control group users were collected. |
| *Chiu et al. (2021)* | A total of 9,458 depressive posts from 260 depressed Instagram users and 22,286 nondepressive posts from 260 nondepressed Instagram users were collected. |
| *Tommasel et al. (2021)* | A Twitter dataset, called SpanishTweetsCovid19, including more than 150 million tweets between March 1st and August 30th 2020 was collected during COVID-19 pandemic in Argentina. |
| *Kim et al. (2020)* | A total of 633,385 Reddit posts from 248,537 users from January 2017 to December 2018 were collected. |

### Stemming and lemmatizing

Stemming is used to reduces a base word to its stem word. For example, the words connect, connected, connecting, connections all can be stemmed to the word "connect" (*Khyani et al., 2021*). In literature, different methods have been used for stemming of words in a text, however, Porter Stemmer and snowball stemming are the widely used ones (*Deshpande & Rao, 2017*; *Kumar, Sharma & Arora, 2019*; *Du, Bian & Prosperi, 2019*; *Geetha et al., 2020*). On the other hand, Lemmatizing is the process of grouping inflected forms of a word to analyse it as a single item. For lemmatizing words, the authors have mostly used Natural Language Toolkit (NLTK)'s Word-NetLemmatizer (*Yazdavar et al., 2017*; *Resnik et al., 2015*) (NLTK is a useful tool for pre-processing and sentimental analysis of text data (*Horecki & Mazurkiewicz, 2015*) function.

### Preprocessing summary

The purpose of preprocessing is to clean up the raw data from noise and the other useless texts such as stop words, non ACSII characters, and punctuation. Furthermore, make the text simpler by reducing similar words to their same base. Although, preprocessing is an essential step to make text simple and suitable to processes it through the classification algorithms, however, it is a time consuming process and consumes most of the time of the researchers that may be used to build efficient algorithms. Furthermore, if a person with no domain knowledge preprocesses the data, there may be a chance of information loss (*Shah et al., 2022a*).

After preprocessing of text, the next step is to extract important features from it for the further processing.

**Table 5** Table depicts the training features containing term frequencies of each word in each document.

|       | He | is | going | home | today | not | hope | can | go | tomorrow |
|-------|----|----|-------|------|-------|-----|------|-----|----|----------|
| $X_1$ | 1  | 1  | 1     | 1    | 1     | 0   | 0    | 0   | 0  | 0        |
| $X_2$ | 1  | 1  | 1     | 1    | 1     | 1   | 0    | 0   | 0  | 0        |
| $X_3$ | 2  | 0  | 0     | 1    | 0     | 1   | 1    | 1   | 1  | 1        |

## Features extraction

Extracting features from a text is an important step of text classification applications. The aim of features extraction is to encode preprocessed text in a form that is ready to train the classifier and minimizes information loss. In the NLP literature two different approaches have been used to extract features from the text *i.e.,* vectorization based features extraction methods like BoW (*Zhang, Jin & Zhou, 2010*), N-gram (*Cavnar & Trenkle, 1994*), and TF-IDF (*Ricardo Baeza-Yates, 1999*) and embedding based methods such as Word2vec (*Mikolov et al., 2013*), GloVe (*Pennington, Socher & Manning, 2014*), and FastText (*Bojanowski et al., 2017*). Detail of the important features extraction techniques related to mental illness literature is provided below.

### Bag of words

BoW is a simplistic representation used in natural language processing. In this method, the text is represented as a bag (different groups) of its words, without the use of grammar, however, maintaining a multiplicity. In principle, BoW establishes a vocabulary of all the unique words of the document ignoring the word order. For example, suppose we have these 3 documents:

1. $X_1$-"He is going home today"
2. $X_2$-"He is not going home today"
3. $X_3$-"He hope he can go home tomorrow"

BoW will create a vocabulary by listing all the unique words of the documents *i.e.,* {'He' 'is' 'going' 'home' 'today' 'not' 'hope' 'can' 'go' 'tomorrow'}. After creating the vocabulary list, the frequency of each word is inserted as shown in Table 5.

BoW has been proved to be a useful features extraction technique for mental illness classification using text data (*Deshpande & Rao, 2017*; *Kim et al., 2020*; *Oliveira & Paraboni, 2024*), however, because of its high dimensions (equal to the total vocabulary size in the document), it may overfit the underlying model. To avoid this, some well-known dimensionality reduction techniques such as PCA, Factor Analysis (FA) *etc.* in combination with BoW may be used.

### N-gram

Like BoW, N-gram also builds a dictionary of words, however, it considers the information of adjacent words in order to construct the dictionary. By counting and recording the occurrence frequency of all fragments, the probability of a sentence can be calculated using the frequency of relevant fragments in the record.

Different mental illness related studies like *Ameer et al. (2022)* and *Safa, Bayat & Moghtader (2022)* have adopted N-gram and achieved promising results. For a small size words set, N-gram is fast and agile method (*Wang, 2006*), however, it cannot capture long-range dependencies as present in long sentence structures.

### Term frequency–inverse document frequency

TF-IDF is a famous technique to convert the stream of a text into numeric data. It is one of the widely used method for weighting the text to extract the information. Its work is based on adjusting the frequencies of the words in the text. Frequency is related to the number that each word occurs in the document. The word occurring more frequently assigned a relatively high frequency value as compared to the word occurring less frequently. TF-IDF assigns weight for each word based on the score given by the following formula (*Hussain et al., 2020b*):

$$frequency(t,d,D) = \frac{tf(t,d)}{df(t,d)} \tag{1}$$

where $t$ represents the term of a text in a document $d$. While $tf$ represents term frequency and $df$ refers to the document frequency.

Because of the ease of use, TF-IDF has been the choice of many researchers to use it to build their models. *Vasha et al. (2023)* used TF-IDF in their work along with different ML algorithms. *Dalal, Jain & Dave (2023)* used TF-IDF along with different ML and DL algorithms to compare their performance on depression detection. In their study, the SVM outperformed the other used algorithms. TF-IDF is a useful and an easy to implement method, however, its dependency on BoW methodology makes it a lexical level feature (*De Choudhury et al., 2013*) deficient in capturing semantics as compared to topic modelling and word embedding.

### Word2vec

In Word2vec, words are represented by dense vectors where a vector represents the projection of the word into a continuous vector space. The position of a word within the vector space is learned from text and is based on the words that surround the word when it is used. The position of a word in the learned vector space is referred to as its embedding (*Shah et al., 2020*). Word2vec has been extensively used in mental illness related studies. *Banna et al. (2023)* used Word2vec along with DL based models for depression detection task. *Baghdadi et al. (2022)* used Word2vec along with TL architectures for the depression detection task.

Word2vec handles different NLP problems effectively, however, its biggest challenge is not recognizing the words which are not the part of the training dataset. Even in large training datasets, some words appear rarely and cannot be mapped to vectors during training. Furthermore, using the softmax function with BoW, the model training becomes very difficult because of high number of categories (large vocabulary size). Approximation algorithms like negative sampling and hierarchical softmax have been proposed in literature to address these issues (*Rong, 2014*).

### Global vectors for word representation

Like Word2vec, GloVe also treat words as the atomic unit to train on, however, GloVe focuses on words co-occurrences over the whole corpus (*Ramírez-Cifuentes et al., 2021*). Its embedding relates the probabilities of two words appearing together. GloVe ensures that the frequency of co-occurrences is an important information and should not be wasted. It builds word embedding in a way that a combination of word vectors relates directly to the probability of these words' co-occurrence in the corpus.

In the mental illness detection literature, GloVe is a popular technique and has been extensively used. *Alabdulkreem (2021)*, *Baghdadi et al. (2022)* used GloVe embedding along with LSTM and BERT and achieved promising results in predicting depression from social media posts of users.

GloVe is an important technique, however, sometimes, as compared to the other embedding methods, it produces undesirable results. For example, *Kancharapu & Ayyagari (2023)* used GloVe, Word2vec, FastText, and their weighted average fusion *i.e.,* GloVe+Word2vec+FastText along with CNN, LSTM, BiLSTM, and Gated Recurrent Units (GRUs). In their study, GloVe achieved least performance as compared to the other used embedding methods.

### FastText

FastText embedding is actually an extension of the Word2vec embedding. It improves Word2vec embedding model by taking word parts into account. This trick enables training of embedding on smaller datasets and generalization to unknown words. It treats each word as composed of character N-gram, so the vector for a word is made of the sum of it (*Shah et al., 2020*).

Like Word2vec and GloVe, many recent studies have adopted FastText and achieved remarkable results. *Fudholi (2024)* used FastText along with CNN-BiLSTM model to detect multiple mental illness conditions. Theair model achieved over all 85.0% accuracy in detecting different mental illness conditions. In another recent study, *Tejaswini, Sathya Babu & Sahoo (2024)* used FastText embedding along with CNN and LSTM based hybrid classifier to detect depression from users social media data and achieved the promising results as compared to the state of the art.

### Topic modelling

Topic modeling is a statistical modeling technique used to discover abstract topics in a document/collection of documents. It is a convenient method to organize, understand, and summarize a large collections of textual information. In this review, a widely used topics modeling technique *i.e.,* latent dirichlet allocation (LDA) (*Blei, Ng & Jordan, 2003*) is focused. LDA is based on the assumption that a mixture of topics creates a document. These topics then generate words based on their distribution of probability. Many researchers have used LDA in their research to detect mental illness. *Tong et al. (2022)* used LDA along with CBPT for the task of depression detection through social media data and achieved a considerable detection accuracy. *Ramírez-Cifuentes, Mayans & Freire (2018)* used LDA and LIWC to detect anorexia nervosa form users social media posts. They found that using LIWC and LDA Features, improves the result of their model.

The topic modeling is an important NLP technique, however, it experiences a significant performance loss when used for short texts analysis. It is because it lacks the word co-occurrence information in each short text and does not produce desirable results (*Albalawi, Yeap & Benyoucef, 2020*).

## Classification methods

After features extraction, the extracted features data is used to predict a class or label for classification of possible mental illness conditions. Classification is a procedure to map a function $f$ between input variables $x$ and the output variable $Y$ *i.e.,* $Y = f(x)$. In literature, different ML and DL methods have been used for mental illness detection and classification (refer to 'Extensive Literature Review' for detail on different employed ML and DL methods). This section describes the popular ML and DL methods that have been employed for mental illness classification using social media data.

### Traditional machine learning methods
*Support vector machine*

SVM is a supervised ML algorithm used for classification and regression application tasks (*Jakkula, 2006*). In the classification applications (like the mental illness classification), it classifies data by finding an optimal hyperplane that maximizes its margin between each class in an N-dimensional (N represents the number of features) space (*Khan et al., 2021*).

According to Fig. 2, there may exist different hyperplanes to distinguish the two groups of data points, however, the goal of the SVM is to find a hyperplane with the maximum margin (*Noble, 2006*). SVM has been extensively used in literature for the classification of different mental illness conditions. *Vasha et al. (2023)* compared the performance of SVM with different ML algorithms. In their study, SVM achieved the highest accuracy in detecting depression. *Dalal, Jain & Dave (2023)* compared the performance of SVM with the DL algorithms and achieved the highest accuracy in detecting depression from the users social media data. *Ramírez-Cifuentes, Mayans & Freire (2018)* explored four ML algorithms *i.e.,* LR, RF, SVM, and MLP for detecting anorexia nervosa. In their study, SVM achieved the highest detection accuracy.

SVM has always been the first choice of the researchers to use when there is no any prior information about the data. Also, the SVM works well with the unstructured and semi-structured data. Furthermore, the regularisation capability of SVM allows it to avoid over-fitting the model. With all such capabilities, still, like the other ML algorithms, SVM suffers from a few limitations. For example, SVM is not a right choice to be implemented with large datasets. In such cases, it takes longer time to train. It also faces difficulty in understanding variable weights and the interpretation of the final model. *Fatima et al. (2018)* used SVM and RF to classify depressive and non-depressive classes of the users on social media. In their study, RF outperformed the SVM. *Deshpande & Rao (2017)* compared SVM with Multimodal Naive Bayes (MNB) for emotion detection. In their study, MNB outperformed the SVM. *Singh et al. (2020)* used five different ML algorithms to detect antisocial behaviour on social media. In their study, the SVM with count vectorization showed the best results, however, RF along with TF-IDF vectorization outperformed the other employed algorithms.

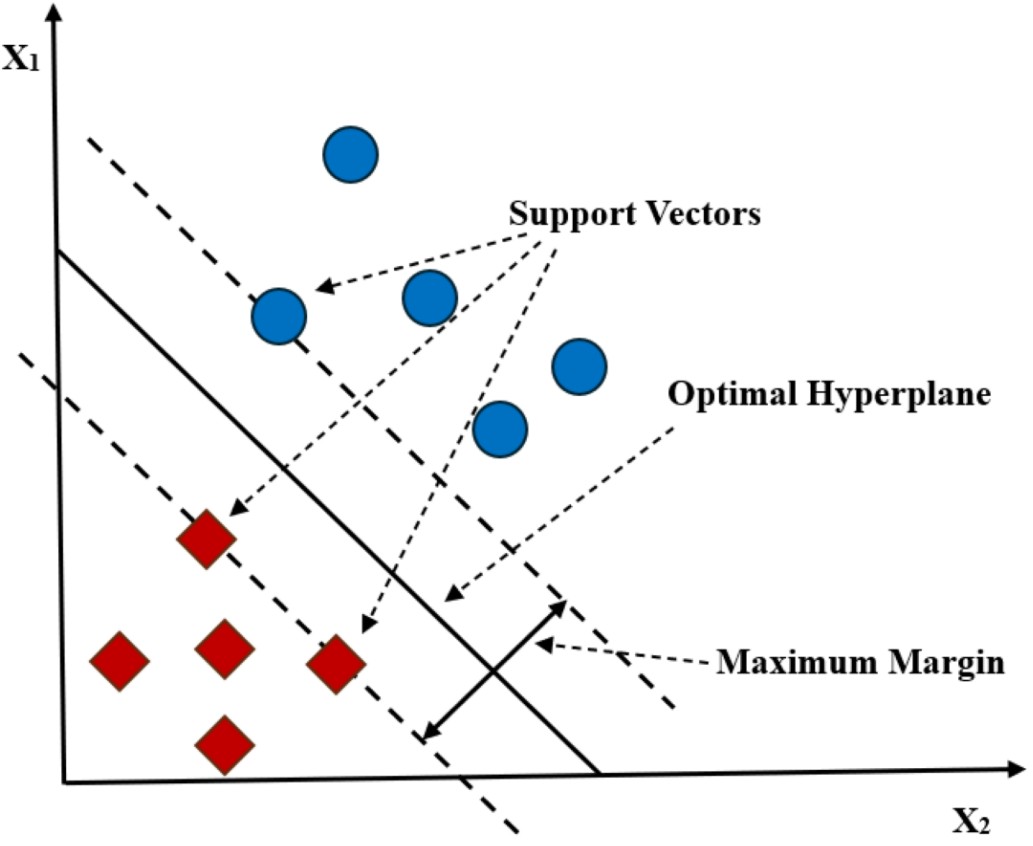

**Figure 2** **Classifying objects with a support vector machine.**

### *Logistic regression*

LR is a statistical ML technique that has been widely used for different text classification applications like mental illness detection using social media data. LR uses logistic function to achieve classification tasks *i.e.,*

$$Sigmoid(x) = \frac{1}{1 + e^{-x}}. \tag{2}$$

LR is one of the most used classification algorithms and has been extensively used for mental illness detection tasks. *Lin et al. (2017)* presented their research to detect stress from users posts. In their research, LR showed the best detection performance. The authors of *Alqazzaz et al. (2023)* used different conventional ML algorithms *i.e.,* LR, KNN, SVM, along with CNN-LSTM classifier for depression detection. *Vasha et al. (2023)* also used LR along with the other ML algorithms for depression detection task on Facebook and YouTube data.

LR can be effectively used with linear data problems, however, it is not a right choice for non-linear problems. In such cases, it can be outperformed by more complex models. For example, *Ramírez-Cifuentes, Mayans & Freire (2018)* compared performance of four ML algorithms including LR to detected anorexia nervosa from users social media data. They found that LR falls behind SVM.

### Naive Bayes

NB is a probabilistic algorithms based on Bayes' theorem with the "naive" assumption of conditional independence between every pair of features (*Saritas & Yasar, 2019*). Bayes' theorem can be mathematically expressed as:

$$P(A|B) = \frac{P(A)P(B|A)}{P(B)} \tag{3}$$

where $P(A|B)$ is the probability of the occurrence of event $A$ when event $B$ has occurred, $P(A)$ is the probability of the occurrence of $A$, $P(B|A)$ is the probability of the occurrence of event $B$ when event $A$ has occurred, $P(B)$ is the probability of the occurrence of $B$. NB is a popular choice of the researchers to use it for mental illness classification problems. *Deshpande & Rao (2017)* compared the performance of NB and SVM on the text-based emotion detection applications using Twitter data. In their system, NB outperformed SVM. *Kumar, Sharma & Arora (2019)* proposed a hybrid model based on MNB, Gradient Boosting (GB) and RF to predict anxious depression disorder in real-time tweets. The proposed model achieved a considerable accuracy in detecting depressed users on Twitter. *Govindasamy & Palanichamy (2021)* compared the performance of NB with a hybrid NB model *i.e.,* NB Tree in depression detection using Twitter data. In their study, both the algorithms performed equally well and provided the same accuracy levels. of *Samanvitha et al. (2021)* compared the performance of NB with the LR, RF, and SVM classifiers for detecting depression using users' social media text data. NB performed better as compared to the others in detecting depression.

Although, in many studies NB has been successfully used for mental illness detection using users social media data, still in some studies it did not perform well as compared to the other classifiers. For example, *Singh et al. (2020)* observed in their work that NB provided less accuracy as compared to SVM and RF. In the study of *Simms et al. (2017)* NB also failed against LR in detecting cognitive distortions.

## Deep learning methods
### Multilayer perception

MLP is composed of the layers of perceptrons (*Tang, Deng & Huang, 2015*; *Shah, Memon & Salam, 2020*) as shown in Fig. 3.

A perceptron is a simple unit that calculates the sum of product of its inputs and their corresponding weights and finally applies a non linear function on it to calculate its output (*Shah et al., 2017*) Mathematically,

$$\hat{y} = f\left(\sum(\mathbf{WX}) + b\right) \tag{4}$$

where, $\hat{y}$ is the predicted output (in case of supervised ML problem), $W$ and $X$ represent weight and feature vectors, respectively, and $b$ denotes bias, which is a single scalar number. Perceptron is also the name of an early algorithm for binary classification based on supervised learning. Because of the simplicity of the architecture, MLP has been the choice of many researchers of the mental illness detection and classification field. *Ramírez-Cifuentes, Mayans & Freire (2018)* used MLP for anorexia detection on social media data and achieved considerable detection accuracy. *Lakhotia & Bresson (2018)* compared the

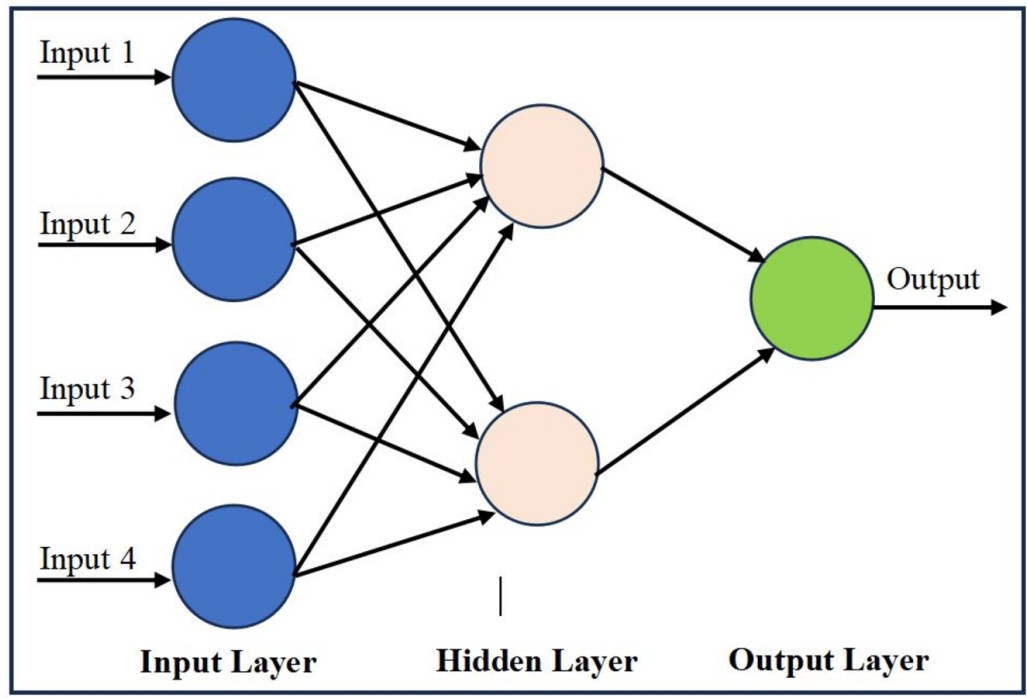

**Figure 3** A simple MLP architecture.

performance of SVM, MLP, and DL models and their fusion for the mental illness detection through Twitter data. In their study, MLP outperformed the other applied algorithms. *Saranya & Kavitha (2022)* proposed an improved MLP algorithm for the early detection of mental illness disorder that handles the voluminous dataset based on maximal relevancy and minimal redundancy. Comparing with the other popular ML algorithms like DT, SVM, and the traditional MLP, their proposed approach achieved the highest accuracy in predicting mental disorders. *Javed et al. (2021)* used MLP and SVM to predict the risk of depression and anxiety in pregnant women. They trained their model on a dataset of 500 Pakistani women. Their proposed classifiers achieved Area Under Receiving Operating Characteristic Curve *i.e.,* AUC scores of 88.0% and 80.0% for antenatal depression and 85.0% and 77.0% for antenatal anxiety, respectively.

Although, MLP has been the choice of the many studies, however, some times it could not be the best choice for text classification. It is because the MLP is a fully connected network and in case of the large network it often provides the undesirable results. For example, *Mohamed et al. (2023)* employed MLP, SVM, and RF classifiers in order to early predict Anxiety problems. In their study, RF achieved the highest accuracy as compared to MLP.

### Convolutional neural network

Convolutional neural network (CNN) is a type feed forward neural network that is able to extract features from the input data with convolution structures different from the traditional feature extraction methods (*Li et al., 2021*; *Shah et al., 2021*). CNN is comprised

of three types of layers *i.e.,* convolutional layers, pooling layers, and fully-connected layers. When these layers are stacked, a CNN architecture is formed (*O'Shea & Nash, 2015*; *Sajid et al., 2023*) as shown in Fig. 4. Along with the other multiple applications, CNN is a popular architecture for mental illness detection and has been the choice of the many researchers of this field. *Gamaarachchige & Inkpen (2019)* proposed a multi-channel CNN with three different kernel sizes for the task of mental illness detection using social media text. Their proposed system achieved 80.0% detection accuracy. *Agarwal et al. (2023)* implemented ensemble DL model using CNN and RNN (see next subsection for detail on RNN) for detecting different mental illnesses diseases such as anxiety, bipolar, dementia, and psychotic. Based on the parameters *i.e.,* accuracy, precision, recall, and F1-score, the proposed ensemble model performed better as compared to the existing models. *Kim et al. (2020)* employed XGBoost and CNN for the task of mental illness detection using users posts from social media *i.e.,* Reddit. In their model, the highest accuracy was achieved by CNN. *Hassan, Hussain & Qaisar (2023)* used the fusion of CNN and different ML algorithms in order to identify Schizophrenia. Their hybrid model achieved over all 98.0% identification accuracy in identifying Schizophrenia. *Fudholi (2024)* employed a CNN-BiLSTM model aided by a FastText-based word weighting method to detect BPD, anxiety, depression, bipolar, schizophrenia, and poison. The dataset used was containing 35,000 training records and 6,108 test records. The employed model yielded an F1-score and accuracy of 85.0% and 85.0%, respectively. *Gorai & Shaw (2024)* proposed a DL model to identify suicide risk using a combination of BERT and an ensemble of multiple CNN. BERT was used for encoding the text data extracted from the social media posts into numerical representations to capture the context-aware meaning of words and phrases in the text, while ensemble CNN was used to analyze the encoded text data to identify patterns relevant to suicide risk. The model was trained on a large corpus of text data from social media and suicide notes. The results showed that the proposed model performed better as compared to the recent approaches in detecting suicide risk. *Tejaswini, Sathya Babu & Sahoo (2024)* proposed a FastText-CNN and LSTM for depression detection from users' social media posts. The proposed model used FastText embedding for better text representation along with semantic information, a CNN to extract global information, and an LSTM to extract local features with dependencies. The proposed model achieved an overall 88.0% accuracy to detect depression. Comparative results with the other state of the art methods showed a considerable improvement in accuracy using the proposed model.

Although, CNN has produced remarkable results in mental illness detection, however, it requires large amount of data to train (*Vázquez-Abad et al., 2020*). Therefore, it is not preferable for the specific applications where the dataset is small. Additionally, the performance of CNN is highly dependent on the quality of data. In case of noisy data, the models learn and recognize noises rather than the original data. Furthermore, implementing CNN may be computationally expensive and a time consuming process, which is not required in many cases (*Krichen, 2023*).

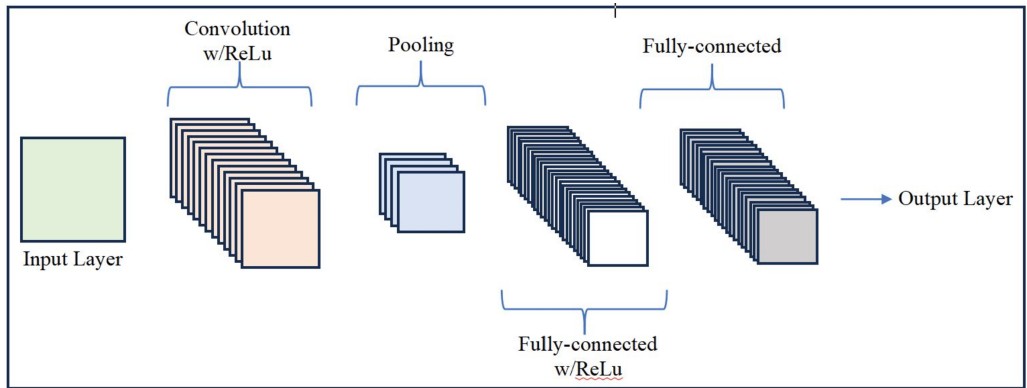

**Figure 4** **A simple CNN architecture.**

### Recurrent neural network

RNN is a neural network with feedback connections. The simplest form of fully recurrent neural network is an MLP with the previous set of hidden units activation feeding back into the network along with the inputs as shown in Fig. 5. RNN is capable of learning features and long term dependencies from sequential and time-series data. LSTM is a type of RNN (*Graves, 2012*) that uses loops to extend the memory of RNN (*Katte, 2018*). For a sequence data, RNN and LSTM are excellent tools. The power of RNN and LSTM for sequence modeling attracted researchers to use them for mental illness detection applications. *Cong et al. (2018)* used BiLSTM (putting two independent LSTMs together) for depression detection. They used XGBoost to reduce imbalance in their data taken from Reddit in form of Self Reported Depression Diagnosis (RSDD). Their model based on BiLSTM significantly outperformed the previous state of the art models on the RSDD dataset. *Ranganathan et al. (2019)* used RNN to detect anorexia from users posts extracted from Reddit. They compared the results of RNN with SVM and demonstrated that the RNN outperformed the SVM. *Bouarara (2021)* used RNN to detect different mental disorders like anxiety, phobia, depression, paranoia by analyzing their social media data through Twitter. The experimental results demonstrated that RNN provides the best results *i.e.,* 85.0% of accuracy as compared to the other techniques in literature such as social cockroaches, DT, and NB. *Apoorva et al. (2022)* compared the performance of simple RNN and LSTM to detect depressive users through their tweets. In their study, the LSTM model outperformed the RNN by having a validation accuracy of 96.2% and a validation loss of 0.1077. *Kanahuati-Ceballos & Valdivia (2024)* evaluated three different models *i.e.,* RNN with LSTM, RNN with BiLSTM for detecting depressive comments in social media. The results showed that the optimized model achieved a precision of 83.3%.

Like the CNN, RNN has also been a choice of many researchers of text classification research, however, it may not be the best choice for the applications where limited computation power and small memory space is available to process the data. It is because the RNN implementation requires a high computational capability and a large memory space to process data (*Rezk et al., 2020*). RNN also lacks in building highly accurate, robust,

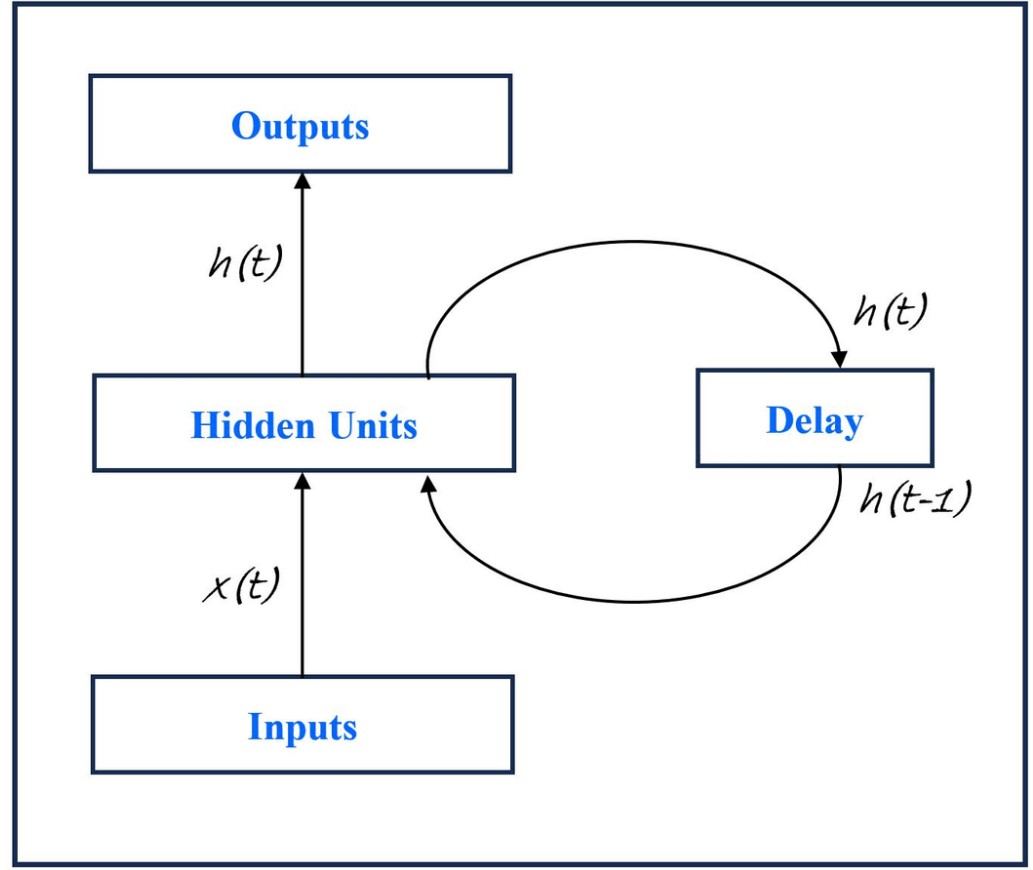

**Figure 5** A simple fully connected RNN architecture (*Salman et al., 2018*).

and efficient models (*Chen et al., 2023*) and needs improvements in its implementation procedures for non expert users. Because of this reason users prefer other easy to use statistical models like Eata-squared, and Auto-regressive Integrated Moving Average (ARIMA) over traditional RNNs (*Hewamalage, Bergmeir & Bandara, 2021*).

## CURRENT CHALLENGES AND FUTURE DIRECTIONS

Detecting mental illness through social media data is not an easy task. It has several challenges to overcome.

First and the foremost issue is the data availability. The relevant and the right amount of data is always a difficult task (*Saravia et al., 2016*). *Deshpande & Rao (2017)* collected data through crawling tweets that contain depression keywords. They could collect only a limited amount of data. Because of training with the limited amount of data, their model suffered from the lack of precision. *De Choudhury et al. (2013)* also faced a similar issue. In their research, they used a limited amount of data to train the classifier and hence, their trained model could not perform up to the mark. The study presented in *Kumar, Sharma & Arora (2019)* was again limited because of the small amount of relevant data.

Various other reviewed studies such as *Dalal, Jain & Dave (2023)*, *Amanat et al. (2022)*, *Almars (2022)* and *Bae, Shim & Lee (2021)* faced the same issue of the limited size and the quality of the datasets. *Zhang et al. (2023)* reported in their study that the most of the available datasets for the mental illness detection are private and only limited datasets are publicly available. Furthermore, the available datasets are not properly annotated and have annotations biases. All such limitations related to the available datasets limit the number and scope of studies related to mental illness detection.

Another issue is the diversity in the users' behaviours on social media. Because of this reason it is extremely difficult to find mental illness related features to cope with the mental health issues. Such limited features issue was faced by many researchers in literature. For example, *Wang et al. (2017)* and *Fatima et al. (2018)* could get only a few features to detect people with eating disorders (ED). The work of *Simms et al. (2017)* also faced the same issue. To cope with this issue, many recent studies, along with mental illness related posts, have collected other than mental illness related posts like fitness, jokes, meditation, parenting, relationships, and teaching *etc.* (*Bae, Shim & Lee, 2021*). Some recent studies (*Wani et al., 2022*; *Chiu et al., 2021*) have used hybrid features.

Features extraction methods also play an essential role in text classification applications. Various models in literature used LDA (topic modelling) as feature extractor (*Kotenko, Sharma & Branitskiy, 2021*; *Dao et al., 2017*; *Shen et al., 2017*; *Resnik et al., 2015*; *Tsugawa et al., 2015*; *Yazdavar et al., 2017*; *Ramírez-Cifuentes, Mayans & Freire, 2018*). Traditional topic modelling is not able to approximate semantic understanding of short texts (*i.e.*, tweets) very well; therefore, researchers have investigated some other well known NLP based feature extraction techniques like BoW, N-gram, TF-IDF, Word2vec, GloVe, and FastText *etc.* (*Oliveira & Paraboni, 2024*; *Ameer et al., 2022*; *Amanat et al., 2022*; *Banna et al., 2023*; *Baghdadi et al., 2022*; *Fudholi, 2024*). Since every feature extraction technique possesses some limitations (refer to 'Features extraction' for more detail on the limitations of features extraction techniques), researchers have investigated other methods for effective textual feature extraction. For example, many recent studies have employed attention mechanism (*Ahmed, Lin & Srivastava, 2023*; *Ji et al., 2022*), and knowledge graphs (*Cao, Zhang & Feng, 2022*). Feature enrichment and data augmentation techniques (*Shams & Jabbari, 2024*) have also been used to cope with the limited features issue. Moreover, multi model features like image, text, and audio (*Meshram & Rambola, 2023*; *Park & Moon, 2022*; *Zhang et al., 2020*) have been used to provide comparatively better performance than a single text based technique.

The choice of the classification models is also a challenging issue. Many studies have employed traditional ML algorithms like LR, NB, KNN, and DT (*Alqazzaz et al., 2023*; *Vasha et al., 2023*; *Dalal, Jain & Dave, 2023*; *Ameer et al., 2022*; *Bae, Shim & Lee, 2021*; *Kim et al., 2020*). Most of these algorithms were used in supervised manner. The advantage of using these algorithms in supervised mode is to achieve better performance with limited amount of labeled training data. However, manually labeling data is a time consuming and prone to error process (*Shah et al., 2021*). Furthermore, these methods heavily rely on manual feature engineering for optimal performance (*Su et al., 2020*). For this purpose, many recent studies have opted DL algorithms (*Oliveira & Paraboni, 2024*; *Gorai & Shaw,*

*2024*; *Fudholi, 2024*; *Banna et al., 2023*; *Amanat et al., 2022*; *Wani et al., 2022*; *Figuerêdo, Maia & Calumby, 2022*; *Baghdadi et al., 2022*; *Almars, 2022*; *Alabdulkreem, 2021*). DL approaches are based on an end-to-end mechanism to map the input raw data directly to the output without the need of manual features engineering. DL models have produced interesting results in the mental illness detection field (refer to Table 3 for the detail on the obtained results of DL algorithms); however, their main concern is the amount and quality of the available data. DL models usually require huge amount of data to train and the amount of data required to train DL models is really hard to obtain. Specially from the social media platforms where the data comes from different sources because of different writing styles and semantic heterogeneity.

As future directions, the understanding of the social networks data be focused to enhance sentiment analysis for mental illness detection. A new suggesting layer can be added to the existing neural network layers to decrease the false positives cases. Other features such as social behavioral, user's profile or time may be used for better detection of mental illness. Furthermore, other DL strategies such as reinforcement learning, multi task learning, and multiple instance learning may be investigated. Hybrid models by combining DL and TL models may also be used for better detection. Researchers of mental illness detection and identification have investigated numerous mental illness conditions through social media. Depression was the most covered mental illness condition that attracted by many researchers. In future, those mental illness conditions can be focused, which are deprived and never been focused before *i.e.,* loneliness, which can be considered one of the early depression symptoms and its early detection can help in early depression detection. Most of the work of the automatic mental illness detection using social media data has been done in the developed countries (US, Europe, *etc.*) and many of the world regions are still out of it. As a future work, more regions can be identified and focused for automatic mental illness detection through their social networks data. The critical analysis of the recent ML and DL models revealed the need of more recent transformer based methods to be explored in this specific area. Hence, this area could be of special interest for the researchers for future research and investigation.

## CONCLUSION

Mental illness is a prevalent disease and a leading cause of disability worldwide. Millions of people suffer from mental disorders; however, only a very small fraction receive a proper treatment. The major reason is its difficult traditional diagnostic procedures. Luckily, the social media has proven to be a valuable source that can provide important data for mental illness detection and classification. In this review study, the recent literature related to mental illness detection, through social media data has been extensively reviewed. The following are contributions of the study:

- Discussion on different data collection methods to collect data from social media platforms.
- Discussion on data prepossessing techniques to preprocess the collected data from social media platforms.

- Discussion on popular features extraction techniques to extract features from the collected data.
- Discussion on popular classification techniques including ML and DL models to detect and classify different mental illness conditions.

This review covers more than 100 recent related literature, along with their contributions and limitations. Furthermore, every part of the review is critically discussed with recent relevant references. It is explored that the data collection related to mental illness from social media platforms is one of the biggest challenges. It is because the available data on social media can be unstructured, irregular and confused due to the misspelling in writing or social media abbreviations during posts. This contributes in different types of uncertainties in language, lexical, syntactic, and semantic. Hence, it is more challenging to evaluate and extract knowledge patterns from such datasets. Bag of Words, TF-IDF, Word2vec, GloVe, FastText, and topic modeling were found to be the popular feature extraction techniques to extract features from the users text data. Along with detail of these feature extraction techniques, their limitations have been discussed in detail. Different ML and DL classifiers employed for mental illness detection and classification, have been discussed and their critical analysis is presented. It was found that SVM, NB, DTs, LR, *etc.* are some of the well known traditional ML methods that have been widely used for mental health detection and classification among the others. Furthermore, CNN, RNN, and their different variants were also found the well known methods employed for mental illness detection and classification. Twitter and Reddit were the biggest social networks that were used as data collection platforms. Meanwhile, depression was the most chosen mental illness condition by the researchers to investigate.

This current literature review provides a complete road map on how mental illness detection models can be designed using social media data and ML and DL models. It provides detailed information about the important preprocessing, features extraction, and classification techniques that have been frequently employed in this specific area. Eventually, this work provides the readers with the directions on how state of art systems in this specific area can be designed in future.

### Funding
The authors received no funding for this work.

### Competing Interests
The authors declare there are no competing interests.

### Author Contributions
- Shahid Munir Shah conceived and designed the experiments, performed the experiments, analyzed the data, performed the computation work, prepared figures and/or tables, authored or reviewed drafts of the article, and approved the final draft.

- Mahmoud Mohammad Aljawarneh performed the experiments, prepared figures and/or tables, and approved the final draft.
- Muhammad Aamer Saleem performed the experiments, prepared figures and/or tables, and approved the final draft.
- Mahmoud Saleh Jawarneh performed the experiments, prepared figures and/or tables, and approved the final draft.

## Data Availability

This is a literature review.

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
