# Peer review of "Mental illness detection through harvesting social media: a comprehensive literature review"

_PeerJ Computer Science, doi:10.7717/peerj-cs.2296_

## Round 0.1 · original submission · Major Revisions

Based on the reviewer comments, the manuscript must be revised before acceptance.

**Language Note:** The review process has identified that the English language must be improved. PeerJ can provide language editing services - please contact us at [email protected] for pricing (be sure to provide your manuscript number and title). Alternatively, you should make your own arrangements to improve the language quality and provide details in your response letter. – PeerJ Staff

Reviewer 1 ·

Basic reporting

The manuscript entitled “Mental illness detection through harvesting social media: A comprehensive literature review” has been investigated in detail. The paper aims to explore the identification of mental illnesses, particularly depression, through analysis of users' social media data. It discusses various feature extraction, detection, and classification techniques, focusing mainly on Deep Learning (DL) models. However, the paper lacks specificity in methodology and critical analysis of the literature. Despite reviewing numerous DL, ML, and NLP models, it fails to provide detailed insights into their technical contributions. Moreover, the absence of empirical evidence and concrete performance metrics undermines the credibility of the study. Overall, the paper falls short in providing a comprehensive roadmap for identifying mental illnesses using social media data. There are some points that need further clarification and improvement:
1) The introduction fails to provide a clear and concise problem statement. While it highlights the significance of mental illness, it lacks specificity regarding the research gap or the novelty of the study.
2) The assertion that mental illness is "increasingly plagued" in modern society needs empirical evidence to support it. Without concrete data, such claims are speculative and lack credibility.
3) The methodology section is deficient in detailing the specific techniques employed for feature extraction, detection, and classification from social media data. It lacks depth in explaining the algorithms and models used.
4) The paper mentions a comparison between traditional Machine Learning (ML) and Deep Learning (DL) approaches but fails to provide critical insights or analysis into the strengths and weaknesses of each approach.

Experimental design

The literature review lacks depth and critical analysis. While it claims to present the most recent literature, it doesn't offer a nuanced understanding of the current state-of-the-art methodologies or their limitations.

The review of over 100 DL, ML, and NLP models is commendable but lacks clarity in categorizing and analyzing their technical contributions and strengths.

Validity of the findings

The paper mentions a quantitative analysis of DL models on popular benchmarks but fails to provide specific details or results of this analysis.

Without concrete performance metrics or comparative evaluations, the paper lacks empirical evidence to support its claims of effectiveness in identifying mental illnesses using social media data.

The conclusion is weak and fails to summarize the key findings or contributions of the study effectively.

The presentation of future research directions lacks specificity and fails to provide actionable insights for further investigation.

The authors should clearly emphasize the contribution of the study. Please note that the up-to-date of references will contribute to the up-to-date of your manuscript. The study named- “A comparative analysis of artificial intelligence optimization algorithms for the selection of entropy-based features in the early detection of epileptic seizures”- can be used to explain the methodology in the study or to indicate the contribution in the “Introduction” section.

It will be helpful to the readers if some discussions about insight of the main results are added as Remarks.

This study may be proposed for publication if it is addressed in the specified problems.

Additional comments

The manuscript entitled “Mental illness detection through harvesting social media: A comprehensive literature review” has been investigated in detail. The paper aims to explore the identification of mental illnesses, particularly depression, through analysis of users' social media data. It discusses various feature extraction, detection, and classification techniques, focusing mainly on Deep Learning (DL) models. However, the paper lacks specificity in methodology and critical analysis of the literature. Despite reviewing numerous DL, ML, and NLP models, it fails to provide detailed insights into their technical contributions. Moreover, the absence of empirical evidence and concrete performance metrics undermines the credibility of the study. Overall, the paper falls short in providing a comprehensive roadmap for identifying mental illnesses using social media data. There are some points that need further clarification and improvement:
1) The introduction fails to provide a clear and concise problem statement. While it highlights the significance of mental illness, it lacks specificity regarding the research gap or the novelty of the study.
2) The assertion that mental illness is "increasingly plagued" in modern society needs empirical evidence to support it. Without concrete data, such claims are speculative and lack credibility.
3) The methodology section is deficient in detailing the specific techniques employed for feature extraction, detection, and classification from social media data. It lacks depth in explaining the algorithms and models used.
4) The paper mentions a comparison between traditional Machine Learning (ML) and Deep Learning (DL) approaches but fails to provide critical insights or analysis into the strengths and weaknesses of each approach.
5) The literature review lacks depth and critical analysis. While it claims to present the most recent literature, it doesn't offer a nuanced understanding of the current state-of-the-art methodologies or their limitations.
6) The review of over 100 DL, ML, and NLP models is commendable but lacks clarity in categorizing and analyzing their technical contributions and strengths.
7) The paper mentions a quantitative analysis of DL models on popular benchmarks but fails to provide specific details or results of this analysis.
8) Without concrete performance metrics or comparative evaluations, the paper lacks empirical evidence to support its claims of effectiveness in identifying mental illnesses using social media data.
9) The conclusion is weak and fails to summarize the key findings or contributions of the study effectively.
10) The presentation of future research directions lacks specificity and fails to provide actionable insights for further investigation.
11) The authors should clearly emphasize the contribution of the study. Please note that the up-to-date of references will contribute to the up-to-date of your manuscript. The study named- “A comparative analysis of artificial intelligence optimization algorithms for the selection of entropy-based features in the early detection of epileptic seizures”- can be used to explain the methodology in the study or to indicate the contribution in the “Introduction” section.
12) It will be helpful to the readers if some discussions about insight of the main results are added as Remarks.
This study may be proposed for publication if it is addressed in the specified problems.

·

Basic reporting

References are presented in the text without curly brackets. I would suggest using them for the sake of clarity.

The abstract clearly presents the scope of the paper but misses on the clear goal that is to perform a systematic review. Consider a paragraph to present the data sources and the research questions.

On table 1, the reference year is duplicated, please remove. Reformat the table to be more pleasant to read, compressing the size. I would suggest ditching the bullets on the contributions, this seems to introduce unnecessary white spaces. What is the difference between data sources and data (heading of column 3)? What databases, criteria and search phrases were used to obtain this table? Where any of the resulting references eliminated? Consider PRISMA approach for this. Replicate the table heading on all pages.
Survey methodology should be placed before the results. So, section 3 should precede section 2.
Section 2 naming is not convincing to me. I would merge section 3 and 2.

Experimental design

Section 4 starts with a conclusion form table 1 that is, in my view, not attainable. What part of table 1 presents the two approaches for data collection? More than referring that “is adopted by many of the reviewed studies” I would like to know what studies. This section is rather confusing, with some works taking several phrases and other just one or two. I would also divide the surveyed works in paragraphs, not starting a new work on the same paragraph as the previous one. This would help steer the reader’s attention.

Correct the typo on section 4.1 “platfomrs”  platforms. Also, on the second phrase of section 4.2 “aouthors”  authors. Although well written in general, the paper needs careful editing.
Authors provide a table with the most common datasets. It would be interesting to know what the datasets contain: a simple ground truth labeling as depression/no depression or different levels? What features do these datasets focus on?

So up to section 4 authors presented a table of papers related to the subject, some data about the search process and datasets. However, no information about how the process itself is done, although there are some references to tweets. Can we extrapolate that the data to be analyzed is text or image? And what features are extracted from such data types? Section 5 begins with a rather strange approach that lacks reference – authors point to issues with data analysis. Are these issues real in the current case? How? In what papers do authors found such problems? I miss the presentation if the global picture here, before being presented with data analysis approaches. From 5.1 it seems text is the data item to be analyzed…

Table 4 focus on 9 papers, authors refer 100. What happens to the others? No data about preprocessing is given? No preprocessing is done? What are the advantages and disadvantages of text preprocessing?
Data analysis is a section of the paper, but focuses on 9 papers, giving some information about the different approaches but no data about performance or suitability.
Typo in line 239 – calculate instead of calculates

Validity of the findings

Authors provide many examples related to the topic, but also digress by presenting papers that process EEG signals.

“A word document with ten occurrences is more relevant than a term frequency document but it is not ten times more relevant”. Please clearly explain the entities present in this phrase, if possible using an example.
Clearly explain the equations in section 6.0.2. In equation 1 what is t and d? Is log the same as le (Napierian log?). On equation 2 what is N, d, ft,?

Line 275 – do you mean in instead of at?

Convolutional neural network examples typically use EEG on the examples given, not social media text. Consider adding some works related to social media text or maybe remove his section.

Line 636 – eary should be early
I fail to see the relevant of line 662 contents.

Additional comments

The paper is very big, provides good insights, but could be more centered. Explanations on NN, CNN and RNN are superfluous currently. Digging inside the data sets, explaining how results can be validated and remove off-topic data can greatly improve the paper interest. Moreover, I would also add more information about the results, namely how can they be compared.

Reviewer 3 ·

Basic reporting

This literature review work, entitled "Mental Illness Detection Through Harvesting Social Media: A Comprehensive Literature Review" is very relevant in the current public health and social context.
The authors strived to present a very exhaustive work, presenting text mining and machine learning techniques in context analysis.
However, there are a lot of writing errors in the first part, namely:
- line 64 "Zogan et al. (2022) In literature", missing point;
- From line 86 to 89, the text that describes the sections must be reorganized, going from section 2 to 4 and then returning to 3. Nor should the paragraph end with "....";
- Several acronyms are not defined "EEG", "API", among others, a table of acronyms is suggested;
- In table 1, "BiLSTM" only appears defined in line 26. If the table of acronyms is not presented, then each one must be defined in the first occurrence of the text;
- Line 97 "literature reveiew" should be corrected;
- Line 103 "Detection using Scoicla Media" should be corrected;
- Line 119 "the usrs’ relavent data" should be corrected;
- Line 152 "The aouthors of" should be corrected;
- Figure 1 and 2 have the same caption;
- Line 650 "with the mantel" should be corrected;

Experimental design

No comment.

Validity of the findings

Chapter 8 does not present a coherent discussion with the 22 articles that are present in table 1, with practically all of those mentioned here being prior to 2019. There is no reference in this chapter to the possibility of global confinement due to COVID-19 , whether or not it caused any deterioration in mental health.
Although it is a review article, conclusions that contribute to knowledge must be analyzed and drawn, not just the presentation of techniques or algorithms used. If that is the goal, then it is very unambitious.

The conclusions are too vague without presenting new proposals for lines of investigation based on the work analyzed. The authors state that "It could be useful for researchers and could provide knowledge to them for developing novel classification frameworks, that could be useful to analyze the data in social networks. Eventually, this work has a great potential to be further explored in the future .", in what way? What is the contribution of this review task in terms of aggregating knowledge? What is its usefulness after all for researchers who wish to use it as a reference?

Additional comments

In Acknowledgments:
"So long and thanks for all the fish."
In a scientific article of this nature, this joke has no context and should therefore be removed. If there are any acknowledgments, they must have some formality, referring to the entities or people who in some way contributed to the work.

---

## Round 0.2 · accepted · Accept

Based on the reviewers opinion, the manuscript can be accepted.

Reviewer 1 ·

Basic reporting

All my comments have been thoroughly addressed. It is acceptable in the present form.

Experimental design

All my comments have been thoroughly addressed. It is acceptable in the present form.

Validity of the findings

All my comments have been thoroughly addressed. It is acceptable in the present form.

·

Basic reporting

Authors revised the text taking into account the identified limitations. They also provided some references of the current year.

Experimental design

Authors clearly state the goals and give an overview of the main contributions of the paper in the conclusions.

Validity of the findings

It is a review. Discussion is based on the findings.

Additional comments

I m happy with the new version and do not provide further comments.